# Separation of the Temperature Effect on Structure Responses via LSTM—Particle Filter Method Considering Outlier from Remote Cloud Platforms

**Yang Qin [1], Yingmin Li [1,2,*] and Gang Liu [1]**

1   School of Civil Engineering, Chongqing University, No. 83 Shabei Street, Chongqing 400045, China
2   The Key Laboratory of New Technology for Construction of Cities in Mountain Area of the Ministry of Education, No. 174 Shazheng Street, Chongqing 400044, China
*   Correspondence: liyingmin@cqu.edu.cn

**Abstract:** Structural health monitoring (SHM) has been widely applied in the field of Mechanical and Civil Engineering in recent years. It is very hard to detect damage, however, using the measured data directly from the remote cloud platform of on-site structure, owing to changing environmental conditions. At the same time, outlier data from the remote cloud platform often occurs due to the harsh environmental conditions, interferences in the wireless medium, and the usage of low-quality sensors, which can greatly reduce the accuracy of structural health monitoring. In this paper, a novel temperature compensation method based on a long-short term memory (LSTM) network and the particle filter (PF) is proposed to separate the temperature effect from long-term structural health monitoring data. This method takes LSTMs as the state equation of PF, which solves the problem whereby PF cannot accurately derive the state equation for complex structures. A feedback model using the probability distribution generated by PF is developed to filter the observed value, thus measurement outliers can be successfully reduced. A numerical simulation and the measured deflection data from an SHM system are utilized to verify the proposed method. Results from the numerical simulation show that the LSTM-PF method can satisfactorily compensate for the temperature effect even when the nonlinear temperature effect is considered. Moreover, outputs from the SHM system of a large-scale suspension bridge indicate the temperature effect can be compensated and outliers can be appropriately reduced at the same time using the measured deflection data.

**Keywords:** structural health monitoring; temperature effect; surrogate modeling; outlier elimination; remote cloud platform; sensor

## 1. Introduction

Real-time structural health monitoring (SHM) of full-scale structures has flourished in the last few decades in the mechanical and civil engineering fields [1,2]. The basic principle of SHM is that a number of structural properties such as stiffness and damping are closely related to structural damage [3]. However, the monitoring data are measured from remote cloud platforms during long-term operation conditions, which contain a very large amount of information, including changing environmental conditions, loading and inevitable testing error, and so forth. As a matter of fact, some field tests have found that temperature was one of the significant factors. For example, Farrar [4] performed vibration tests on the I-40 Bridge by cutting one of the girders in four damage levels, and he found that damage cannot be directly detected through the identified frequency because the ambient temperature played a major role in the variation of the bridge's frequency.

To deal with temperature variation, monitoring data or damage features extracted from these data should be compensated or modified to the same environmental conditions. Hence, a great number of methods have been proposed to separate the temperature effect, and they can be divided into two categories: Model-based and model-free methods. When

temperature data are measured, various regression and interpolation methods have been developed based on statistical models [4–7]. Xia has presented the static linear regression algorithm using the measured bridge displacement and temperature to eliminate the temperature effect [5]. H. Sohn used a linear filter between temperature and natural frequency to capture the variation of the frequencies to temperature [6]. Dynamic regression methods such as Auto-Regressive output and an Exogeneous input method (ARX), which can consider the influence of the outputs and inputs at every time instant, have been proposed to separate the temperature effect with high accuracy [7]. If measured data were contaminated with noise, the wavelet transform [8,9] was implemented to reconstruct the data from the interesting frequency range with a high signal-to-noise ratio, and then regression methods were used to separate the temperature effect. When temperature data were unavailable, the underlying relationship between the temperature and the damage-sensitive features was implicitly modeled by singular value decomposition [10], an auto-associative neural network, subspace [11], and co-integration methods, etc. For example, Manson [12] implemented principal component analysis to project the original feature space into a reduced feature space, which is insensitive to temperature, then the damage feature was immune to temperature effect.

Although there are many methods for temperature effect separation or compensation, few of them consider outlier data from remote cloud platforms, which is a common phenomenon in real SHM systems. With the development of wireless transmission and cloud computation, many structural health monitoring systems adopt the 4G or 5G technique to send measured data from sensors to a cloud platform. However, the received data may be corrupted by some factors, such as radio interference and sensor faults, resulting in a false alarm of the monitoring system. Especially in large-scale platforms with so many devices, it is very common to observe sensors injecting corrupt information into the overall system due to these factors [13,14]. If these distorted data are used to train the linear regression model or ARX model or to acquire the reduced feature space, the accuracy of temperature effect compensation will be significantly decreased. Furthermore, nonlinearity between the temperature and structural response is obvious for complex structures, thus compensation precision will decline if the nonlinearity is not handled appropriately. To address these problems, a temperature effect separation method based on long-short term memory (LSTM) and a particle filter (PF) is proposed in this paper.

The rest of this paper is organized as follows. The theories of the PF algorithm and the LSTM method are presented in Section 2; the proposed LSTM-PF combination strategy is developed in Section 3; temperature effect separation cases from a numerical example and a real bridge are discussed in Sections 4 and 5, respectively; and the paper is finally concluded in Section 6.

## 2. Fundamental Principles of PF and LSTM Approach

### 2.1. Particle Filter

If temperature and structural response are both measured, the PF algorithm can be utilized to address the temperature compensation with the prediction strategy. Furthermore, according to the probability distribution generated by the PF calculation, distortion data or outliers can be eliminated by a feedback model.

Firstly, the dynamics model of a structure can be described as

$$\begin{cases} x^{(t)} = f\left(x^{(t-1)}, n^{(t)}\right) \\ y^{(t)} = h\left(x^{(t)}, v^{(t)}\right) \end{cases} \tag{1}$$

where $x$ is state variables, $y$ is the prediction observation data; $n$ is the process noise, which reflects the noise generated by physical factors in the time-varying process; $v$ is the measurement noise, which represents the measured noise of the sensor; the superscript $t$ denotes time step; and $f(\bullet)$ and $h(\bullet)$ are the state transfer function and the observation function, respectively. The state transfer function is generally determined by the object

system and reflects the pattern between the states. In the field of dynamics, motion equations of systems are generally used as the state function. The observation function or equation reflects the relationship between the state value and the observation value.

Theoretically, PF is a Bayesian estimation method based on the Monte Carlo technique, so the sampling importance resampling filter (SIR) [15] is chosen to solve Equation (1). The calculation steps are briefly as follows:

(1) Initialization

At time step $t = 0$, initial sampling particles $x_i^{(0)}$ ($i = 1, \ldots, N$) are generated by the prior probability density function (PDF) $p(x)$, which is often viewed as normally distributed, where $N$ is the total number of particles.

(2) Importance sampling

At the $t$th time step ($t = 1, 2, \ldots$), the state equation $f(\bullet)$ is used to obtain $\widetilde{x}_i^{(t)}$ and then the observation equation is implemented to acquire $\widetilde{y}_i^{(t)}$ ($i = 1, 2, \ldots, N$). Then, the general weight equation can be written as [16]

$$w_i^{(t)} = \eta (2\pi \textstyle\sum)^{-1/2} exp \left\{ -\frac{1}{2} \left( y_m^{(t)} - \widetilde{y}_i^{(t)} \right) \textstyle\sum^{-1} \left( y_m^{(t)} - \widetilde{y}_i^{(t)} \right) \right\} \tag{2}$$

$$\widetilde{w}^{(t)} \left( x_i^{(0:k)} \right) = \frac{w^{(t)} \left( x_i^{(0:t)} \right)}{\sum_{i=1}^{N} w^{(t)} \left( x_i^{(0:t)} \right)} \tag{3}$$

(3) Resampling

Based on the principle that the total number of particles remains unchanged after resampling, a new particle set $x_i^{(t)}$ is obtained by resampling the particle set $\widetilde{x}_i^{(t)}$ with its corresponding weight $\widetilde{\omega}_i^{(t)}$. Therefore, these particles with a larger weight are divided into multiple particles, and the particles with a very small weight are discarded. Each particle in the newly formed particle set has the same weight of $1/N$, which can be described as $\left\{ x_i^{(t)}, \frac{1}{N} \right\}_{i=1}^{N}$.

(4) Output

Using resample points $\left\{ x_i^{(t)}, \frac{1}{N} \right\}_{i=1}^{N}$, the posterior PDF of system state $x$ can be approximately expressed as:

$$P \left( x^{(t)} \middle| y^{(1:t)} \right) \approx \widetilde{P} \left( x^{(t)} \middle| y^{(1:t)} \right) = \frac{1}{N} \sum_{i=1}^{N} \delta \left( x^{(t)} - x_i^{(t)} \right) \tag{4}$$

where $\delta(\bullet)$ is the Dirac delta function. $\widetilde{P} \left( x^{(t)} \middle| y^{(1:t)} \right)$ is the posterior PDF of system state $x$ after resampling. The expected value of the result can be approximately calculated as:

$$E \left( x_i^{(t)} \right) \approx \frac{1}{N} \sum_{i=1}^{N} x_i^{(t)} \tag{5}$$

(5) Return to step 2 and repeat this iteration until the stop criterion is triggered.

A more detailed description and information about the PF algorithm with the SIR strategy can be found in reference [16].

### 2.2. LSTM Neural Network Architecture

In the second step of the PF algorithm, the state equation is used to calculate state variables forward in each time step, so it plays a crucial role in prediction accuracy. Unfortunately, the state equation is usually unavailable for complex structures or in situ structures in strictly physical terms. Since the state equation in the PF algorithm is used to calculate state variables $x_i^{(t)}$ according to state variables $x_i^{(t-1)}$ by Equation (1), which can be understood as a time series prediction. So, the LSTM can be utilized as the surrogate

model for the state equation in PF. On the other hand, the LSTM itself has the capability of simulating the nonlinear phenomenon, which can improve the performance of PF as well.

LSTM is a special kind of recursive neural network (RNN), which solves the information preservation problem [17]. Not only will LSTM learn the information of the present moment, but it will also rely on the previous sequence information.

The special block of LSTM cells is three gates [18], called the input gate, forget gate, and output gate. With this peculiar structure, LSTM can store long-term memories in the memory cell, which can avoid the disappearance or explosion of gradient propagating over time. The forget gate decides what information from the cell state is discarded. The input gate determines what new information is put into the cell state. The output gate calculates output $b_{uo}{}^t$ of the LSTM block. The special block of the LSTM cell is depicted in Figure 1. Depending on the three gates, LSTMs allow us to store and access information over a long period. In Figure 1, $x_{in}{}^t$ is the input of the LSTM block ad $b_h{}^t$ is the working memory (hidden state) of the LSTM block.

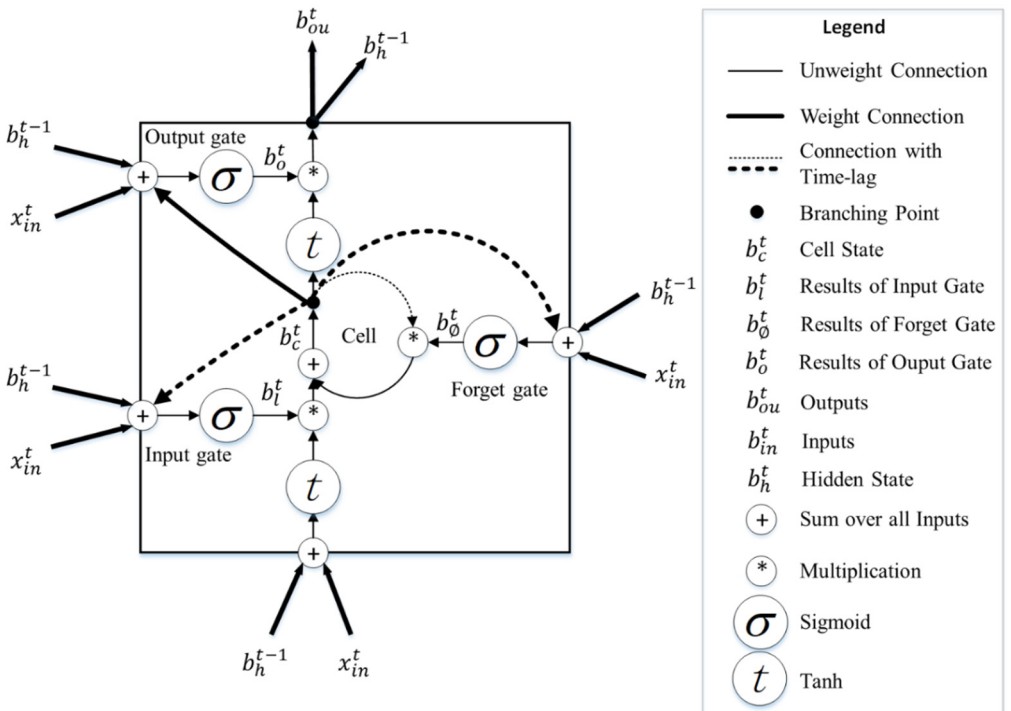

**Figure 1.** LSTM block architecture.

More detailed information about LSTM can be found in reference [18].

### 2.3. Matching of LSTM and State Equation

For complex structures, such as large-span bridges and dams, the equation of motion cannot be obtained directly through theoretical derivation. This problem can be solved if LSTM is used to replace the equation state equation because, firstly, LSTM can learn complex events; secondly, compared with other deep learning methods, LSTM is better at predicting time series, which has the same requirements as the equation of state. Thirdly, LSTM can obtain the state at the next moment from the state at the last moment, which is matched with the state equation. The specific derivation is as follows:

Input gate:

$$a_l^t = \sum_{in=1}^{IN} w_{inl} x_{In}^t + \sum_{c=1}^{C} w_{cl} b_c^{t-1} + \sum_{h=1}^{H} w_{hl} b_h^{t-1} \tag{6}$$

$$b_l^t = f\left(a_l^t\right) \tag{7}$$

Forget gate:

$$a_\varnothing^t = \sum_{in=1}^{IN} w_{in\varnothing} x_{ln}^t + \sum_{c=1}^{C} w_{c\varnothing} b_c^{t-1} + \sum_{h=1}^{H} w_{h\varnothing} b_h^{t-1} \tag{8}$$

$$b_\varnothing^t = f\left(a_\varnothing^t\right) \tag{9}$$

Cell update:

$$a_c^t = \sum_{in=1}^{IN} w_{inc} x_{ln}^t + \sum_{i=1}^{I} w_{hc} b_h^{t-1} \tag{10}$$

$$b_c^t = b_\varnothing^t b_c^{t-1} + b_l^t h\left(a_c^t\right) \tag{11}$$

$$b_c^t = b_c^{t-1} f\left(\sum_{in=1}^{IN} w_{in\varnothing} x_{ln}^t + \sum_{c=1}^{C} w_{c\varnothing} b_c^{t-1} + \sum_{h=1}^{H} w_{h\varnothing} b_h^{t-1}\right) +$$

$$f\left(\sum_{in=1}^{IN} w_{inl} x_{in}^t + \sum_{c=1}^{C} w_{cl} b_c^{t-1} + \sum_{h=1}^{H} w_{hl} b_h^{t-1}\right) h\left(\sum_{i=1}^{I} w_{ic} x_{lstm}^t + \sum_{i=1}^{I} w_{hc} b_h^{t-1}\right) \tag{12}$$

Output gate:

$$a_o^t = \sum_{in=1}^{IN} w_{ino} x_{ln}^t + \sum_{c=1}^{C} w_{co} b_c^{t-1} + \sum_{h=1}^{H} w_{ho} b_h^{t-1} \tag{13}$$

$$b_o^t = f\left(a_o^t\right) \tag{14}$$

Cell output:

$$b_{ou}^t = b_o^t h\left(b_c^t\right) \tag{15}$$

$$b_{ou}^t = f\left(\sum_{in=1}^{IN} w_{ino} x_{ln}^t + \sum_{c=1}^{C} w_{co} b_c^{t-1} + \sum_{h=1}^{H} w_{ho} b_h^{t-1}\right) \times$$

$$h\left(\begin{array}{c} b_c^{t-1} f\left(\sum_{in=1}^{IN} w_{in\varnothing} x_{in}^t + \sum_{c=1}^{C} w_{c\varnothing} s_c^{t-1} + \sum_{h=1}^{H} w_{h\varnothing} b_h^{t-1}\right) + \\ f\left(\sum_{in=1}^{IN} w_{inl} x_{in}^t + \sum_{c=1}^{C} w_{cl} b_c^{t-1} \sum_{h=1}^{H} w_{hl} b_h^{t-1}\right) h\left(\sum_{i=1}^{I} w_{ic} x_i^t + \sum_{i=1}^{I} w_{hc} b_h^{t-1}\right) \end{array}\right) \tag{16}$$

In the case of iteration, the input is $x_i^{t-1}$ and the output is $x_i^t$:

$$x_i^t = f\left(\sum_{i=1}^{I} w_{io} x_i^{t-1} + \sum_{c=1}^{C} w_{co} b_c^{t-1} + \sum_{h=1}^{H} w_{ho} b_h^{t-1}\right) \times$$

$$h\left(\begin{array}{c} b_c^{t-1} f\left(\sum_{i=1}^{I} x_i^{t-1} + \sum_{c=1}^{C} w_{c\varnothing} b_c^{t-1} + \sum_{h=1}^{H} w_{h\varnothing} b_h^{t-1}\right) + \\ f\left(\sum_{i=1}^{I} w_{il} x_i^{t-1} + \sum_{c=1}^{C} w_{cl} b_c^{t-1} + \sum_{h=1}^{H} w_{hl} b_h^{t-1}\right) h\left(\sum_{i=1}^{I} w_{ic} x_i^t - 1 + \sum_{i=1}^{I} w_{hc} b_h^{t-1}\right) \end{array}\right) \tag{17}$$

where the input of LSTM is represented by $x_{in}^t$ and $b_{ou}^{t-1}$ is the output of LSTM. The subscripts $in, l, \varnothing, o, h, c, ou$ represent the inputs, input gate, forget gate, output gate, hidden state, LSTM's cell state, and outputs, respectively. Pooled calculation results are represented by $a$, and $b$ represents activated calculation results. The *sigmoid* activation function is represented by $f$ and $h$ represents the *tanh* activation function.

## 3. The LSTM-PF Algorithm

### 3.1. Environmental Vector

For mechanic and civil structures, the load is assumed to be independent of these environmental factors. Therefore, during the long-term structural health monitoring, the response of the structure $y(t)$ can be obtained by the following equation:

$$y(t) = LF(t) + EF(t) \tag{18}$$

where $LF(t)$ denotes the structural response caused by loads and $EF(t)$ represents the structural response caused by environmental factors.

For in situ civil structures, such as continuous rigid frame bridges, the load has little effect on the structural static response or identified modal parameters, such as natural frequencies, damping ratio, etc. Under this assumption, the $LF(t)$ can be approximately viewed as white noise, so Equation (18) can be rewritten as

$$y(t) = EF(t) + noise1 \tag{19}$$

Furthermore, the effect of temperature is much larger than other operational factors under normal operation conditions. In this regard, other influence factors can also be assumed as white noise

$$EF(t) = TF(t) + noise2 \tag{20}$$

where $TF(t)$ is the structural response caused by temperature. It is worth noting that noise 1 and noise 2 can be viewed as process noise.

It is worth mentioning that wind often has more influence on structures than the temperature effect, but in the code of structural health monitoring, such as the Technical code for the monitoring of building and bridge structures (GB 50982), thee wind is classified as a load, thus it is not considered as an environmental factor in this paper.

To deal with the temperature compensation problem, a temperature vector $T(t)$ is set as

$$T(t) = \left\{ T_x{}^o(t),\ T_x{}^t(t) \right\} \tag{21}$$

where $T_x{}^o(t)$ is the ambient temperature during measurement and $T_x{}^t(t)$ denotes the target temperature for compensation.

*3.2. Temperature Compensation Algorithm*

It can be clearly seen from Equation (17) that LSTM conforms to the iteration rule of the state equation in PF. Moreover, LSTM can handle multiple input problems. Therefore, Equation (1) can be rewritten as

$$\begin{cases} x^{(t)} = LSTM\left( x^{(t-1)},\ T(t) \right) + n^{(t)} \\ \qquad y^{(t)} = x^{(t)} + v^{(t)} \end{cases} \tag{22}$$

It is different from the original PF algorithm in Equation (1) whereby function $f(\bullet)$ is substituted by the LSTM network and the temperature vector $\boldsymbol{T}$ is included.

It should be noted that $y^{(t)} = \left( y_1^{(t)}, y_2^{(t)}, \ldots, y_N^{(t)} \right)^T$ is the compensated observation value at time step $t$. The true measured $y_m{}^{(t)}$, however, includes the temperature effect, thus it cannot be utilized to calculate weights for PF in Equation (3) since they are in a different temperate environment. Fortunately, the trained LSTM network can be used again to acquire the predicted response $y_p{}^{(t)}$ which are under the same temperature environment with the observation $y^{(t)}$. So the compensated predicted response is calculated as

$$y_p{}^{(t)} = LSTM\left( y_m{}^{(t-1)},\ T(t) \right) \tag{23}$$

Since sampling important resampling (SIR) is used in this study, it is only necessary to make a specific choice about the probability density function of the importance of the particle:

$$q\left( x_i^{(t)} \middle| x_i^{(t-1)},\ T(t),\ y_p{}^{(1:t)} \right) = P\left( x_i^{(t)} \middle| x_i^{(t-1)},\ T(t) \right) \tag{24}$$

The weights can be obtained by the following equation:

$$w_i^{(t)} = \eta (2\pi \textstyle\sum)^{-1/2} exp\left\{ -\frac{1}{2} \left( y_p^{(t)} - y_i^{(t)} \right) \overset{-1}{\textstyle\sum} \left( y_p^{(t)} - y_i^{(t)} \right) \right\} \tag{25}$$

After weights are obtained, the normalized weight $\widetilde{\omega}_k^i$ is recomputed according to Equation (4) to obtain the particle set $\left\{ \widetilde{x}_i^{(t)}, \widetilde{w}_i^{(t)} \right\}_{i=1}^N$. The new particle set $\left\{ x_i^{(t)}, \frac{1}{N} \right\}_{i=1}^N$ is obtained by resampling. Then the distribution of particles $\Omega^{(t)}$ is generated from the new particle set $\left\{ x_i^{(t)}, \frac{1}{N} \right\}_{i=1}^N$. Finally, a feedback model, which will be described in Section 3.3, of the output versus the input can be formed by using the distribution $\Omega^{(t)}$. The expected value $E\left( x_i^{(t)} \right)$ can be approximately calculated by Equation (5).

If the temperature and structural response are both measured, the PF algorithm can be utilized to address the temperature compensation with the prediction strategy. Furthermore, according to the probability distribution generated by PF calculation, distortion data or outliers can be eliminated by the feedback model.

### 3.3. Feedback for Eliminating Measurement Outliers from Remote Cloud Platforms

For a real structure, distortion or outlier data are often encountered. Therefore, the $y_m^{(t-1)}$ will mislead the predicted value in the next time step $t$ in Equation (24) and then the whole PF procedure. In this study, the feedback step of PF is used to modify $y_m^{(t)}$, thus the misleading effect of distortion is reduced.

The probability density function $g(x)^{(t)}$ can be acquired from the distribution $\Omega^{(t)}$. Then the $P\left( x^{(t)} \right)$ can be obtained by,

$$\begin{cases} P\left( x^{(t)} \right) = \int_{-\infty}^{y_m^{(t)}} g(x)^{(t)} dx, \ y_m^{(t)} \leq E\left( x_i^{(t)} \right) \\ P\left( x^{(t)} \right) = 1 - \int_{-\infty}^{y_m^{(t)}} g(x)^{(t)} dx, \ y_m^{(t)} > E\left( x_i^{(t)} \right) \end{cases} \tag{26}$$

In the particle filter algorithm, the probability density distribution is composed of discrete particles, and the weight of each particle is equal, so $\int_{-\infty}^{y_m^{(t)}} g(x)^{(t)} dx$ can be approximated as $N_m/N$, where $N_m$ is the number of particles when $y_i^{(t)} \leq y_m^{(t)}$. Therefore, Equation (16) can be rewritten as

$$\begin{cases} P\left( x^{(t)} \right) \approx \frac{N_i}{N}, \ y_m^{(t)} \leq E\left( x_i^{(t)} \right) \\ P\left( x^{(t)} \right) \approx 1 - \frac{N_i}{N}, \ y_m^{(t)} > E\left( x_i^{(t)} \right) \end{cases} \tag{27}$$

In order to avoid the second type II error that takes wrong observations as correct inputs, a threshold $\xi$ is set to determine whether a correction should be made. Although outliers of measured data often occur, in fact, it is a small probability event that the cloud platform generates outlier values in the normal working state. This is because, in practical engineering, the amount of measurement data is large and the measurement time is long, resulting in outliers often occurring among measurement data. Statistically, for such a small probability event, the probability threshold is generally set as less than 1%. Since abnormal data are not completely discarded in the feedback model, this threshold can be appropriately broadened to 1%, that is, $\xi = 1\%$. What is more, it is unreasonable to completely discard observations, because the value of data is wrong, but the trend may be

right to some extent. Consequently, when $P\left(x^{(t)}\right) \leq \xi$, the measured response $y_m{}^{(t)}$ is modified by

$$
\begin{cases}
\hat{y}_m^{(t)} = y_m^{(t)} P\left(x^{(t)}\right) + E\left(x_i^{(t)}\right)\left[1 - P\left(x^{(t)}\right)\right], & y_m^{(t)} \leq E\left(x_i^{(t)}\right) \\
\hat{y}_m^{(t)} = y_m^{(t)}\left[1 - P\left(x^{(t)}\right)\right] + E\left(x_i^{(t)}\right) P\left(x^{(t)}\right), & y_m^{(t)} > E\left(x_i^{(t)}\right)
\end{cases}
\tag{28}
$$

where $\hat{y}_m^{(t)}$ is the observation data after feedback, which is used to amend the input value $y_m{}^{(t)}$ at the next time step in Equation (23).

It assumes that the malfunction of the sensor occurs intermittently, and the damaged structure will constantly produce abnormal data. The feedback model is automatically disabled when it works continuously 5 times, and at this time, the algorithm will lose its ability to eliminate outliers, thus preserving the information of structural damage.

### 3.4. Computation Procedure

(1) Training phase

Under normal conditions, the LSTM model is trained. The dynamic learning rate LR $= 0.001 \times (0.95 \times \text{epoch})$ is adopted for thee LSTM network, where the epoch is the number of iterations. The training operation will terminate when the loss value *Loss* is stable, where *Loss* is defined by

$$
Loss = \frac{\sum_{i=1}^{N}\left(y_p - y_m\right)^2}{N}
\tag{29}
$$

(2) Temperature compensation phase

(1) Initialization

At the first time step $t = 0$, after the initial values $x^{(0)}$ are measured from the sensors, initial sampling particles $\left\{x_i^{(0)}\right\}_{i=1}^{N}$ are generated through prior PDF $P(x)$. Since accurate process and measurement noise are very difficult to obtain explicitly in practical engineering, both of them are assumed as normally distributed with zero mean and variance $\sigma_0$, where $\sigma_0$ is set as 1% root mean square (RMS) of the measured structure response in this article. Furthermore, the total number of particles $N$ is set as 5000 recommended by references [19,20].

(2) Importance sampling

At the time step $t$th ($t = 1,2, \dots$ ), Equation (22), which uses the LSTM network as the state equation, is used to obtain $y_i^{(t)}$ ($i = 1,2, \dots, N$) from $x_i^{(t-1)}$ and $F_x{}^{(t)}$. Then, $y_p{}^{(t)}$ is obtained by Equation (23). $y_i{}^{(t)}$ and $y_p^{(t)}$ are used to calculate the weights $w_i^{(t)}$ for each particle by Equation (25). After that, the normalized weight $\widetilde{\omega}_k^i$ is re-computed according to Equation (3).

(3) Resampling

The new particle set $\left\{x_i^{(t)}, \frac{1}{N}\right\}_{i=1}^{N}$ without weights is obtained by resampling the particle set $\left\{\widetilde{x}_i^{(t)}, \widetilde{w}_i^{(t)}\right\}_{i=1}^{N}$. After that, the distribution of particles $\Omega^{(t)}$ is generated, and the expected value $E\left(x_i^{(t)}\right)$ can be approximately calculated by Equation (5).

(4) Feedback

According to the distribution $\Omega^{(t)}$, the prediction measurement $\hat{y}_m^{(t)}$ is generated by Equations (27) and (28). If the feedback model works five consecutive times, it fails.

(5) Output

The results after temperature compensation $y_p{}^{(t)}$ are obtained by Equation (23).

(6) We return to step 2 and iterate until the end of the time steps.

## 4. Numerical Example

The proposed LSTM-PF method is verified using a finite element (FE) table under different temperatures in this section.

### 4.1. FE Model

In order to verify the feasibility of LSTM-PF for nonlinear temperature compensation, a contrived non-trivial plate model, shown in Figure 2, is conducted with finite element software under tone burst excitation. The density of the plate material is 2700 kg/m$^3$, and its Poisson ratio is 0.33. The initial elastic modulus is $2.0 \times 10^5$ MPa.

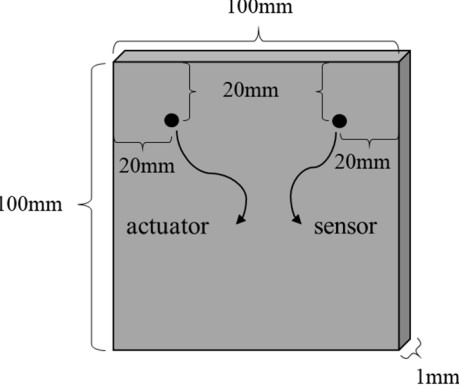

**Figure 2.** The size of the plate.

A 100,000 Hz ultrasonic is used as the input at the actuator and output is measured at the sensor in Figure 2. The ultrasonic is superimposed by a 100,000 Hz sinewave and a Hanning window, as shown in Figure 3.

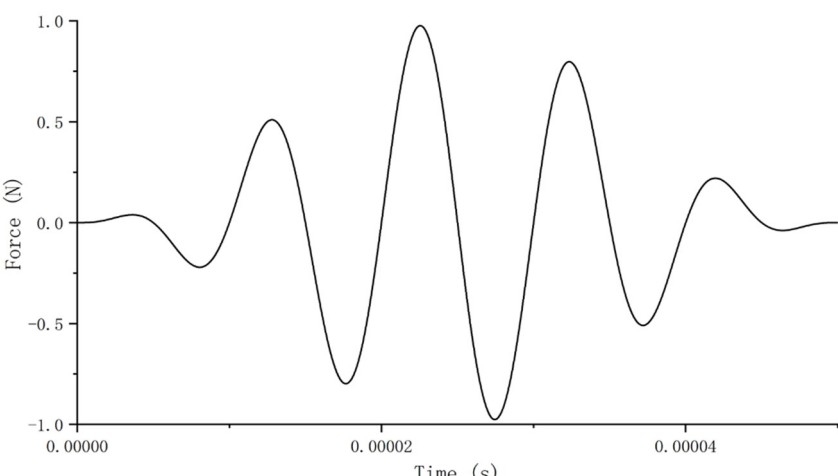

**Figure 3.** The input of ultrasonic.

It is assumed that the elastic modulus of the material varies nonlinearly with temperature, as shown in Figure 4.

Six cases are considered with different temperatures, as listed in Table 1.

**Table 1.** Six temperature cases (units: °C).

| Case | 1 | 2 | 3 | 4 | 5 | 6 |
|---|---|---|---|---|---|---|
| Temperature | 24 | 35 | 47 | 59 | 70 | 80 |

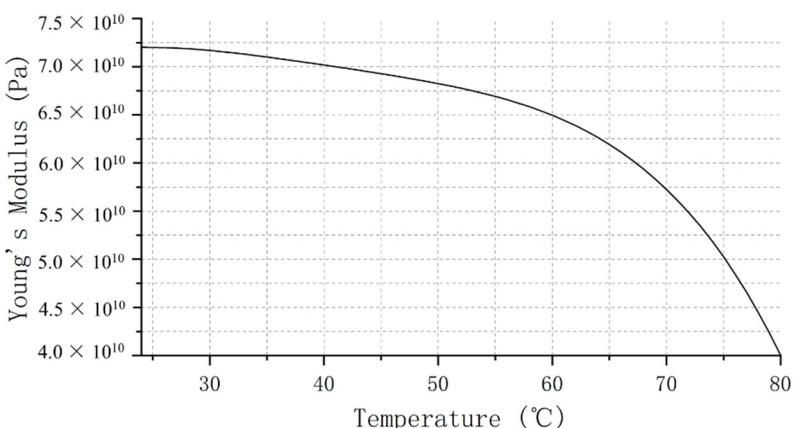

**Figure 4.** The young's modulus vs. temperature of the plate.

In order to simulate the process noise and observation noise in practical engineering, random noise with a signal–noise ratio (SNR) of 60 dB is added for all cases. Furthermore, in order to simulate the corrupt information measured from remote sensing cloud platforms, the outliers with a 1% probability of occurrence and 0~10 random iterations of the RMS of the normal working are added under case 6. The time history of deflection under case 6 with noise and outliers is shown in Figure 5.

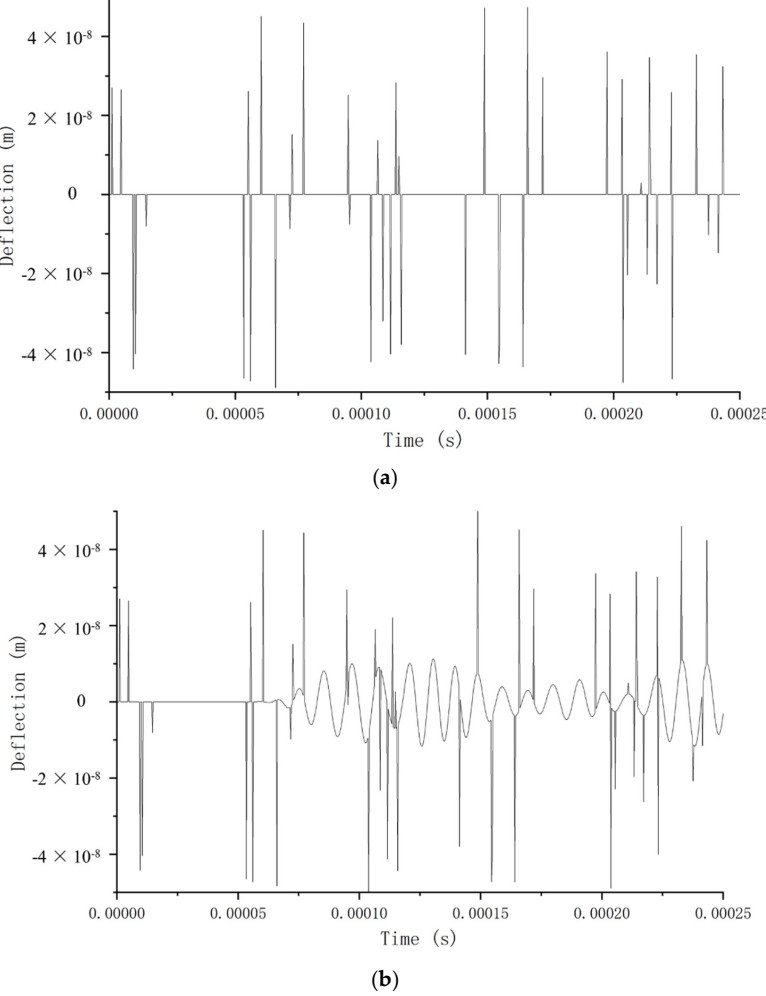

**Figure 5.** The deflection under case 6 with the noise. (**a**) The original outlier signal with noise; (**b**) the combined deflection signal.

### 4.2. LSTM Training and LSTM-PF Model

The LSTM model should be trained before it is combined with PF. The structural response from 0~0.00015 s and its corresponding temperature are used for training, and the subsequent response was regarded as the test set. It is worth noting that six deflections at different cases' temperatures were used for training, which is in line with the conditions of practical engineering, because for real engineering structures, we can obtain structural response data at different temperatures.

For LSTM-PF, the trained LSTM is put into the particle filter to form the new algorithm. After 1000 iterations, stable and good results can be obtained through the LSTM-PF method.

The number of layers in deep learning has a great influence on the accuracy of training. The more layers, the better the ability to extract nonlinear features. However, too many layers will lead to increased training time. The prediction accuracy of the LSTM network with different layers (128 neurons per layer) is shown in Figure 6. The increasing trend of accuracy is obvious before layer 3, but the velocity of increase is extremely slow after layer 3. On the other hand, the training time is prolonged significantly as the number of layers increases (from one to five layers, the consumption time is 25, 32, 43, 57, and 71 min, respectively). Thus, considering the factors of accuracy and computation cost, the number of layers is set to 3 for the LSTM network in this study.

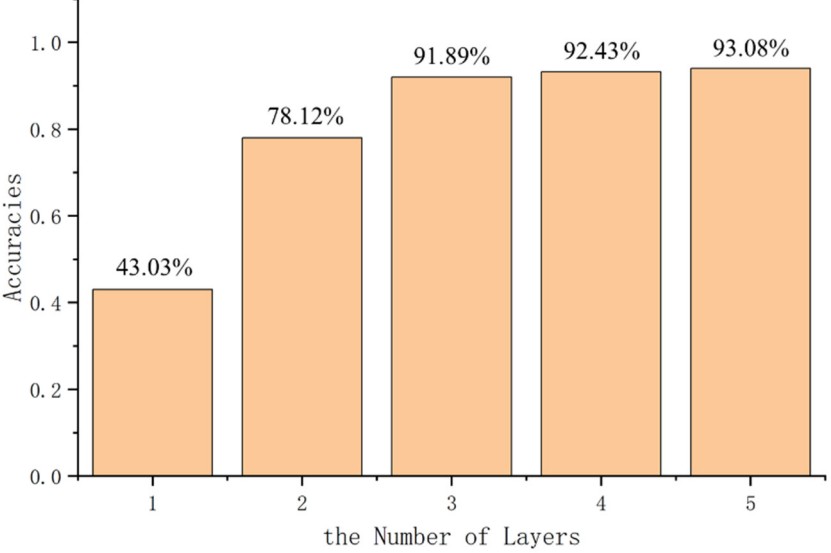

**Figure 6.** Process of LSTM neural network training.

### 4.3. Linear Regression for Temperature Compensation

Linear regression is a traditional statistical method and also a common method to deal with temperature compensation problems. This paper uses linear regression to fit the temperature–stiffness curve. The deflections of the structure are calculated by the finite element method with the stiffness at different temperatures obtained by the fitting.

### 4.4. Results and Discuss

#### 4.4.1. The Signal without Outliers

The deflection under 80 °C is used to compensate for the deflection under 59 °C by the LSTM-PF and LSTM methods as shown in Figure 7.

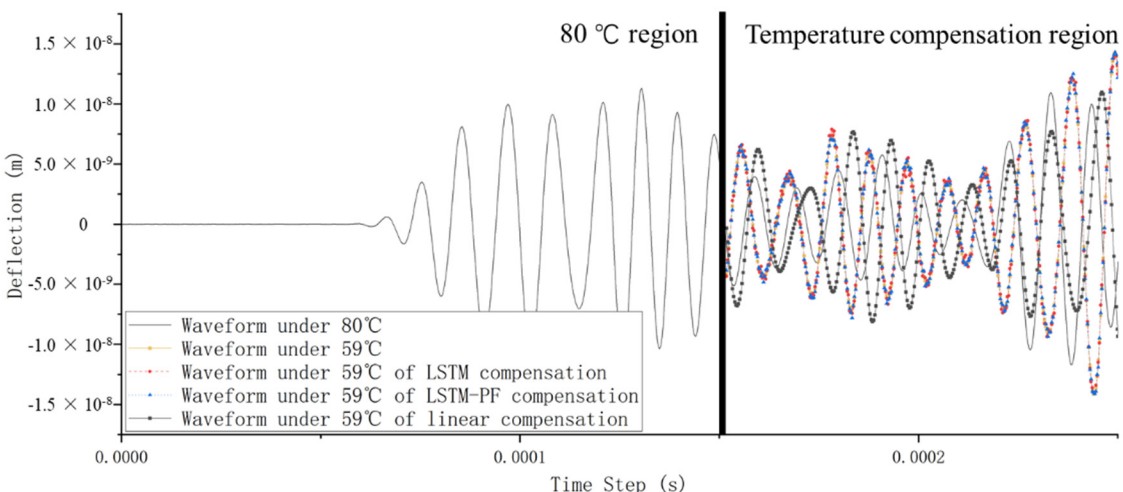

**Figure 7.** The results of compensated defection.

Figure 7 shows that the compensated deflections by LSTM-PF and LSTM coincide with the waveform under 59 °C, which means LSTM-PF and LSTM can perform effective temperature compensation. However, there is a difference between the deformation compensated by the linear regression method and the deflection waveform at 56 °C, indicating that the effect of temperature compensation by the traditional linear regression method is not good. Table 1 shows the mean and variance of relative error. It is worth noting that the relative error here is the ratio of the difference to the RMS value of the reference waveform. This is because there are some points in the wave pattern that are very close to zero. These points close to zero in the denominator tend to make the relative error larger. However, in fact, the absolute error value corresponding to this huge relative error is very small relative to the RMS value of the reference waveform. In order to avoid this situation, the RMS value of the waveform is used instead of the original waveform as the reference value to calculate the relative error.

According to Table 2, the mean and variance of relative error using these two methods are approximately 0.3% and 0.4%. So, LSTM-PF and LSTM methods show good performances on temperature compensation for materials with temperature nonlinearity without outliers. It is worth noting that the mean value of LSTM-PF is slightly less than that of LSTM. This is because the particle filter has a certain filtering effect, so the influence of noise is reduced. However, the mean and variance of relative error using linear regression are approximately 3% and 200%, which are much larger than LSTM and LSTM-PF methods. This shows that LSTM and LSTM-PF methods are better than traditional linear regression methods in dealing with nonlinear temperature effects.

**Table 2.** The relative error without outliers.

| Method | Scenarios | Mean | Variance |
|---|---|---|---|
| LSTM | case 6 compensating case 4 | 0.22% | 0.39% |
| | case 4 compensating case 2 | 0.34% | 0.53% |
| | case 6 compensating case 1 | 0.57% | 0.51% |
| LSTM-PF | case 6 compensating case 4 | 0.12% | 0.21% |
| | case 4 compensating case 2 | 0.35% | 0.42% |
| | case 6 compensating case 1 | 0.32% | 0.23% |
| linear regression | case 6 compensating case 4 | 4.07% | 288.31% |
| | case 4 compensating case 2 | 0.93% | 75.12% |
| | case 6 compensating case 1 | 3.45% | 257.48% |

#### 4.4.2. The Signal Containing Outliers

The deflection under 80 °C with outliers is used as one of the inputs to obtain the compensated deflection under 59 °C (case 6 compensating for case 4). The results of compensation are shown in Figure 8.

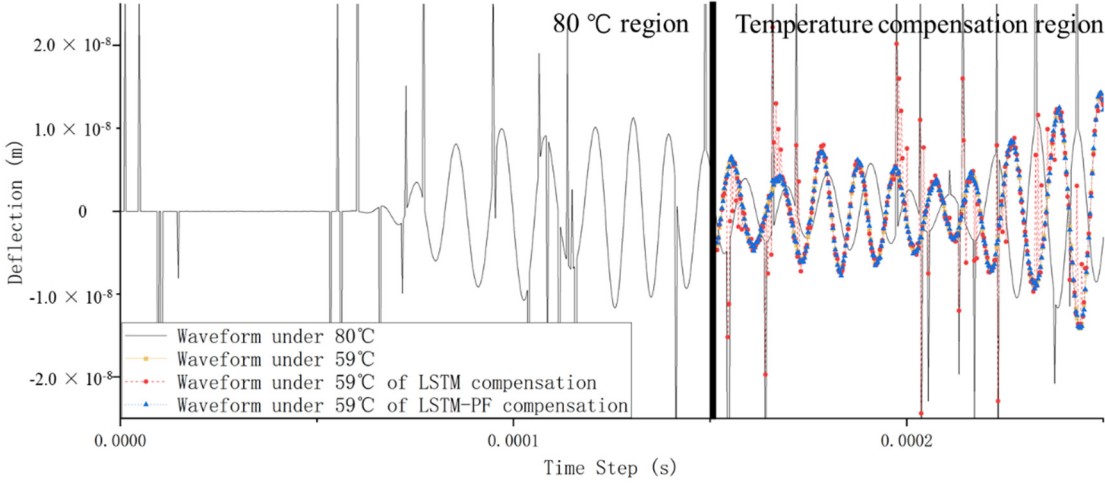

**Figure 8.** The results of compensated deflection with outliers.

Figure 8 shows that the real deflection under 59 °C and the compensated deflection by LSTM-PF also coincide with each other after adding some outliers. Furthermore, there are hardly any outliers in the compensated deflection by LSTM-PF, which means LSTM-PF is good at resisting noise.

Table 3 shows that the relative mean and variance of the proposed LSTM-PF method are 0.13% and 0.22%, respectively. However, the relative mean and variance of the LSTM method increase to 1.55% and 299.39%, respectively. This shows that only the proposed method can compensate for the temperature effect accurately when outliers' values are considered. The reason for these phenomena is that the feedback model of LSTM-PF filter outliers out of the signal so that outliers have no influence on prediction in Equation (12).

**Table 3.** The relative error of results with outlier (case 6 compensating fir case 4).

| Method | Mean | Variance |
|:------:|:----:|:--------:|
| LSTM | 1.55% | 299.38% |
| LSTM-PF | 0.13% | 0.22% |

## 5. Temperature Compensation for a Real Bridge

### 5.1. A Large-Scale Suspension Bridge

The Qiansimen Bridge is a cable-stayed bridge with an 88 m + 312 m + 240 m + 88 m span in Chongqing, China. Figure 9 is a real view of the Qiansimen Bridge. A remote cloud platform is installed in this bridge, and the layout of deflection sensors using a connecting pipe, temperature sensors, and wind sensors is depicted in Figure 10.

The sensors circled in Figure 10a are the ones used in this paper. Firstly, two temperature sensors, QW11 and QW12, are located at the bridge head. Next, the two deflection sensors N11 and N12 are located in the first span. Then, two temperature sensors, QW21 and QW22, as well as a deflection sensor, N21, are in the same position. Finally, the wind sensor is located in the second span.

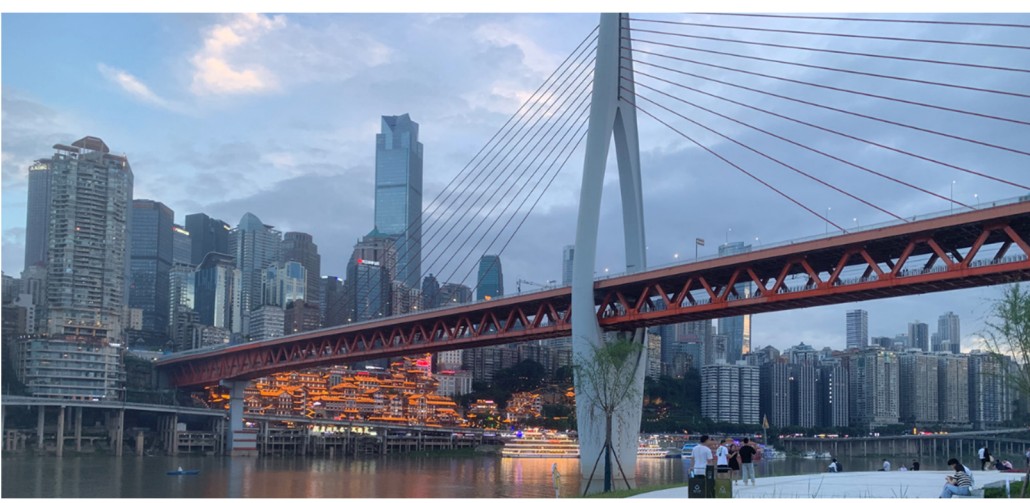

**Figure 9.** The real view of the Qiansimen Bridge.

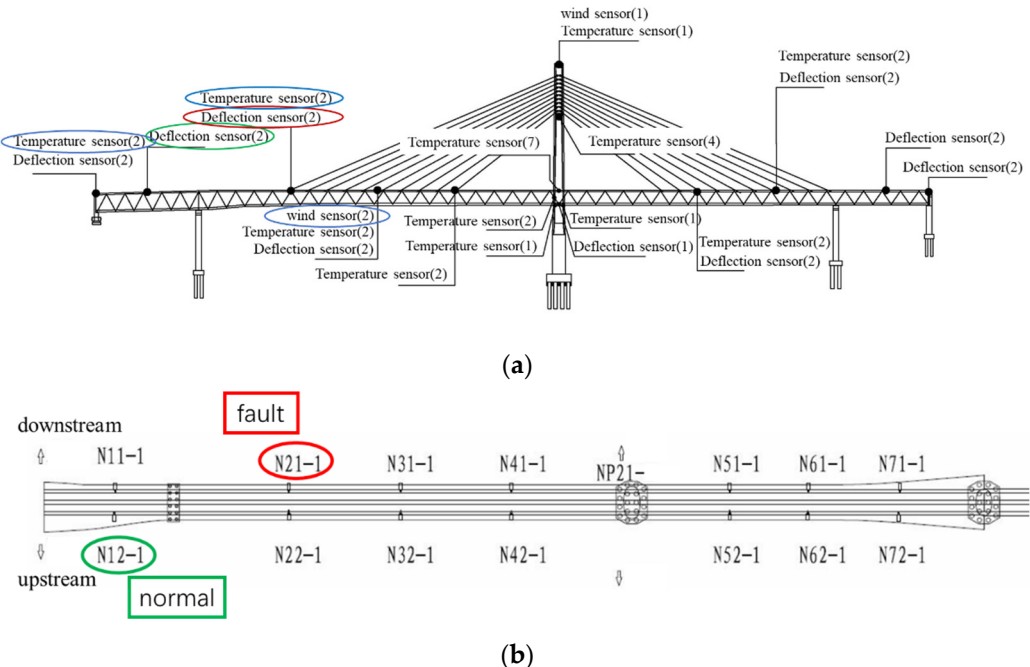

**Figure 10.** Sketch of deflection and temperature sensors' arrangement on the bridge. (**a**) The front view of the bridge; (**b**) The vertical view of the bridge.

The deflection from N12 and temperature from QW21 measured from 2016 to 2017 are shown in Figure 11a. It shows that not only does temperature fluctuate up and down over time, but there are also large seasonal fluctuations for displacement over long periods of time, suggesting that deflection is largely influenced by environmental factors, especially temperature. Figure 11b shows the relationship between temperature and deflection. As can be seen in the figure, it does not exhibit a clear linear relationship, which proves the non-linearity relationship between temperature and deflection. Therefore, if the temperature effect is not separated, the condition assessment of the bridge cannot be implemented accurately.

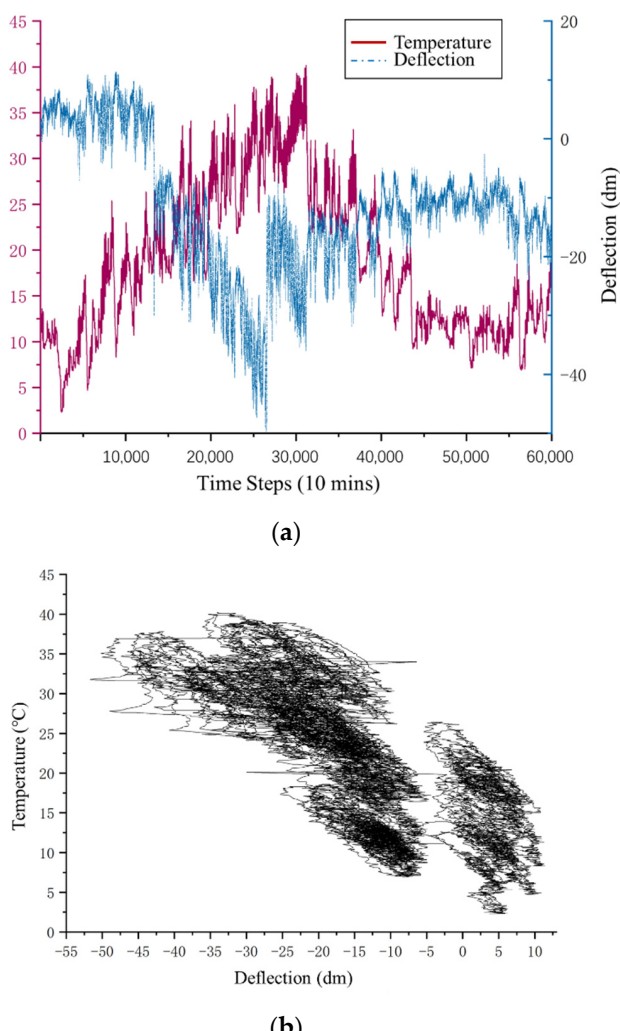

(a)

(b)

**Figure 11.** Temperature changes throughout the year. (**a**) Comparison of temperature and deflection over time; (**b**) Nonlinear relationship between temperature and deflection.

What is more, for the in situ SHM system, outlier data from the remote cloud platform often occurs [21–23]. Figure 12 depicts signals measured by sensors N11, N12, and N21 at the same time. It was observed that data from sensor N11 are consistent with that from sensor N12. The data from sensor N21, nevertheless, are abnormal in the period marked with the dotted box shown in Figure 12.

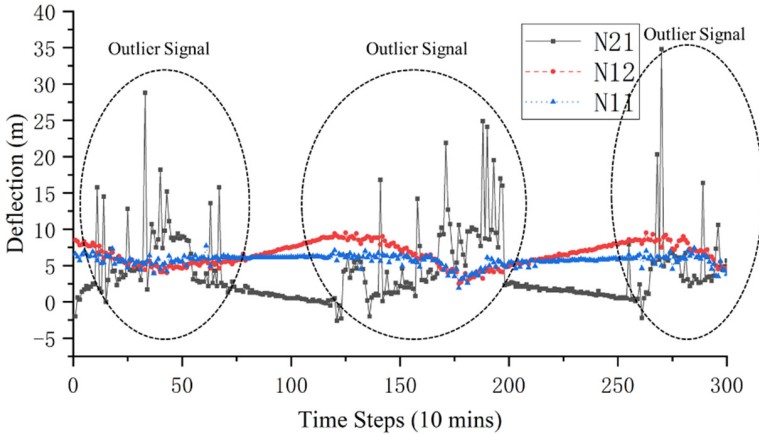

**Figure 12.** Normal data and outliers measured by the sensors.

Figure 13 shows the wind speed measured by the wind sensors QF11-1 and QF11-2 at the same time. It can be seen that when the outlier from the deflection sensor occurs, the wind speed does not show many mutations. That is, the outlier caused by wind load is excluded. It is almost certain that these outliers are caused by some non-structural damage factors, such as harsh environmental conditions, radio interference, and sensor faults.

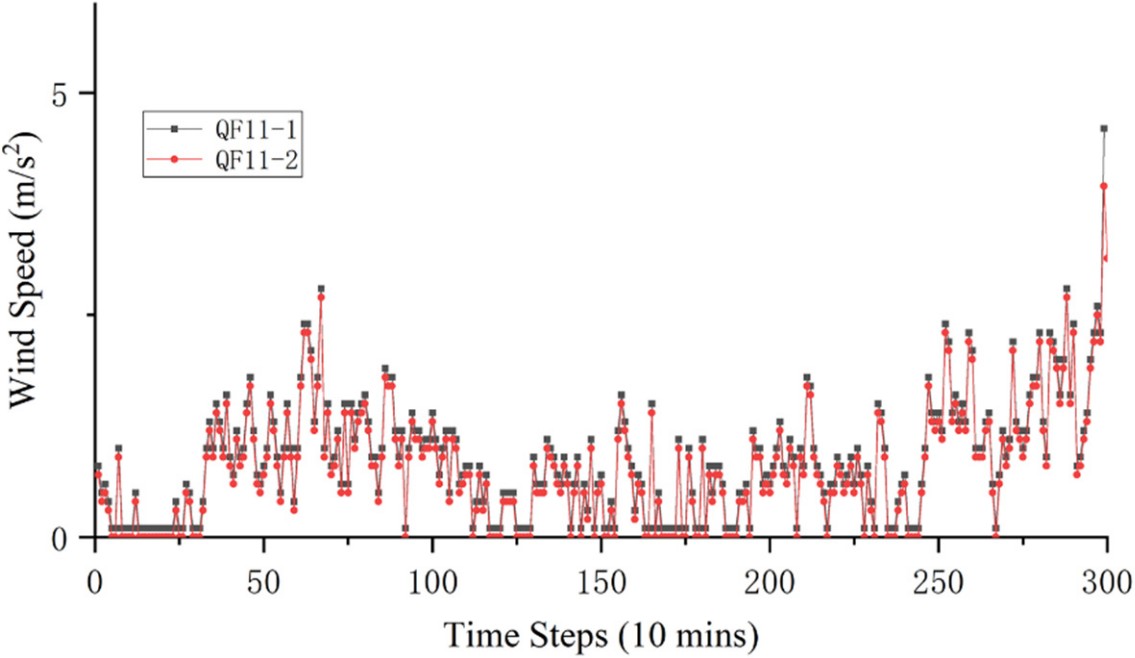

**Figure 13.** Data measured by the wind sensors.

### 5.2. Calculation Model

In order to deal with and without outliers from the remote cloud platform, two cases are considered in this section. Case 1 includes the data measured by temperature sensors QW11, QW12, QW21, and QW22 and deflection sensor N12; Case 2 contains temperature sensors QW11, QW12, QW21, and QW22 and deflection sensor N21.

Since there are many factors that cause the variation in deflection for in situ bridges, the reference or true value after temperature compensation is unavailable. This makes it impossible to choose a base temperature to check whether these methods compensate correctly. However, it is noted that over time, not only the deflection of the bridge changes, but also the environmental measurements such as the temperature of the bridge change, that is, the temperature varies with each time step. So, every prediction at the next moment is actually a calculation of the target value at the next moment under different environmental conditions such as temperature, which can be used as compensation. Therefore, in this example, the prediction can be regarded as compensation, that is, the accuracy of the prediction can be used to prove whether the method can compensate to some extent.

For the training LSTM network, the training data are the deflection data measured by the sensor in normal conditions for one and a half months. The test set is the sensor's deflection data for the next three days.

### 5.3. Results and Comparision

LSTM and LSTM-PF are both used for comparison. The predicted results are shown in Figure 14.

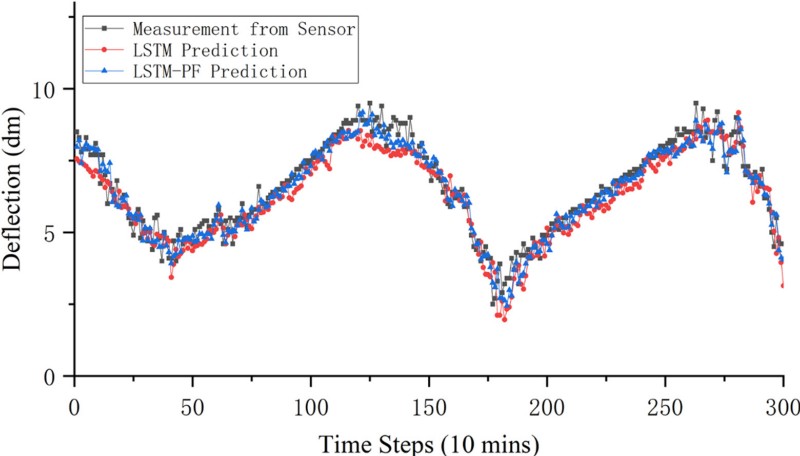

**Figure 14.** Prediction under case 1.

In Figure 14, the solid line (the data measured from the sensor N12-1) is close to the line with the circle (LSTM-PF prediction) and the line with the asterisk (LSTM prediction). It means that the LSTM-PF and LSTM methods can successfully predict the true value.

Table 4 illustrates that the absolute mean error of the predicted value of LSTM-PF and LSTM is only approximately 0.3 dm and 0.5 dm, respectively, whereas the amplitude of the deflection is approximately 3.5 dm, indicating that the absolute mean error is very low. Moreover, Table 4 shows that the mean and variance of relative error between the predicted value calculated by LSTM-PF and the measured data are not bigger than 5%. Therefore, it proves that these two methods can compensate for the temperature of the normal data from the remote cloud platform.

**Table 4.** Error of the prediction without outliers.

| Method | Mean | Variance |
|---|---|---|
| LSTM-PF (absolute) | 0.3081 (dm) | 0.0665 (dm) |
| LSTM-PF (relative) | 4.44% | 2.63% |
| LSTM (absolute) | 0.4907 (dm) | 0.1120 (dm) |
| LSTM (relative) | 7.49% | 4.44% |

Furthermore, the proposed method is used for the abnormal data measured from the remote cloud platform, which can test whether LSTM-PF has the ability to resist outliers in real structures. The results are shown in Figure 15.

Figure 15 shows the predictions under case 2. In the drift sections, the outliers occur frequently. In the normal sections, the signal is no anomaly, which can be used to verify the correctness of the methods.

The line with a circle (LSTM-PF prediction) is very close to the solid line (the abnormal data measured from the sensor N21-1) in the stationary section. It means that the proposed LSTM-PF methods can successfully predict the true value in the stationary section. Furthermore, it fluctuates slightly in the drift section. Although it is impossible to determine whether this prediction is correct, it basically conforms to the basic law of bridge deflection, that is, there will not be a big change in deflection. Therefore, it can be preliminarily judged that the LSTM-PF method is not very sensitive to the outliers from the remote cloud platform and can obtain accurate predictions. However, for the LSTM method, the predicted value fluctuates greatly in the drift section and deviated greatly from the measured value in the stationary section. It can be seen that the LSTM method is very sensitive to outliers.

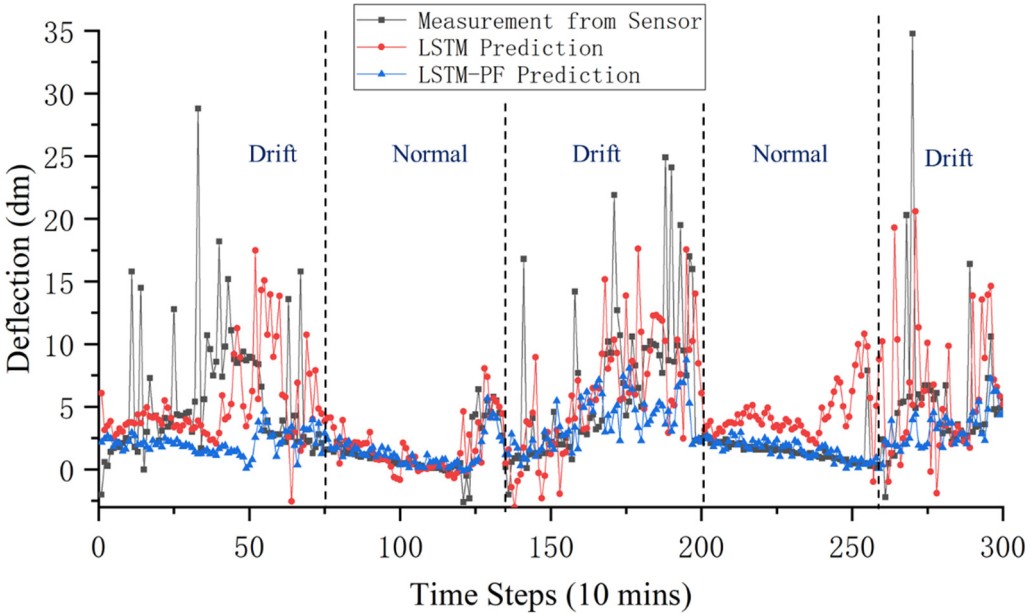

**Figure 15.** Prediction under case 2.

Table 5 shows that the absolute error of the predicted value by the LSTM-PF method is low in the stationary section, which is 0.4170 dm. The variance of LSTM-PF prediction is also small, which is 0.4170 dm. On the other hand, the mean and variance of the predicted error by the LSTM method are 2.4233 dm and 1.9413 dm, respectively, which are approximately 10 times higher than those under a stationary case. It indicates that LSTM is very sensitive to outliers, while outliers have little influence on LSTM-PF. In other words, the LSTM-PF method can compensate for temperature when dealing with the influence brought by singular values well, and its effect is better than that of using the LSTM method.

**Table 5.** Absolute error in normal section.

| Method | Mean | Variance |
|--------|------|----------|
| LSTM | 2.4233 (dm) | 1.9413 (dm) |
| LSTM-PF | 0.4170 (dm) | 0.0760 (dm) |

## 6. Conclusions

A novel method based on LSTM and the Particle Filter for temperature compensation is proposed. This method takes LSTMs as the state equation of the Particle Filter, which solved the problem that PF cannot derive the equation of state for complex structures such as bridges. The feedback model formed by the probability distributions of PF can effectively reduce the negative influence caused by outliers due to harsh environmental conditions, interferences in the wireless medium, and the usage of low-quality sensors. Therefore, this method can deal with the problem of temperature compensation with outliers.

Results from numerical simulation fully illustrate that the single LSTM network is sensitive to outliers. However, the LSTM-PF method can perform temperature compensation and reduce the influence of the outlier after compensation. The outputs from the SHM system of a large-scale suspension bridge prove that LSTM-PF mitigates the impact of the outliers. The proposed method has the advantage that the nonlinear temperature effect can be successfully captured because LSTM is employed as the surrogate for the state equation of the particle filter. Moreover, the feedback model based on the probability distribution generated by PF can filter out the outlier data to improve the accuracy of temperature compensation.

**Author Contributions:** Conceptualization, Y.L., G.L. and Y.Q.; methodology, G.L. and Y.Q.; software, Y.Q.; validation, Y.Q.; formal analysis, Y.Q.; investigation, Y.Q.; resources, Y.Q.; data curation, Y.Q.; writing—original draft preparation, Y.Q.; writing—review and editing, Y.L., G.L. and Y.Q.; visualization, Y.Q.; supervision, Y.L. and G.L. All authors have read and agreed to the published version of the manuscript.

**Funding:** This research was funded by National Natural Science Foundation of China, grant number 51638002.

**Data Availability Statement:** Some data and models during the study are available from the corresponding author by request (1. Simulation model data; 2. measurement data of the Qiansimen Bridge from 2016 to 2017; 3. the results of LSTM-PF in Sections 4 and 5).

**Conflicts of Interest:** The authors declare no conflict of interest. The funders had no role in the design of the study.

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
