# Peer review of "Separation of the Temperature Effect on Structure Responses via LSTM—Particle Filter Method Considering Outlier from Remote Cloud Platforms"

_remotesensing, doi:10.3390/rs14184629_

Round 1

Reviewer 1 Report (Previous Reviewer 3)

This version is a significant improvement over the original submission. It reads well but English editing is needed, especially with regard to the new text.

I suggest that the authors stress the novelty of the proposed method in the conclusions. To this effect, I suggest that the lines 78 to 82 be transferred to the conclusions section. The last sentence in the conclusions can be deleted.

Figure 11 makes sense because we expect downward deflection with the rise in ambient air temperature on a suspension bridge. It would be perhaps more informative to re-plot this relationship to demonstrate non-linearity.

Author Response

Reviewer 2 Report (New Reviewer)

Dear Editor, in my opinion the paper is well written, and the topic falls within the scope of the "Remote Sensing" scientific international journal. Then, in my opinion the paper can be accepted for publication after a minor revision of the english language.
Sincerely,
Rocco Ditommaso

Author Response

This manuscript is a resubmission of an earlier submission. The following is a list of the peer review reports and author responses from that submission.

Round 1

Reviewer 1 Report

The Structural health monitoring is important for key point bridge. Two new dectection prediction metnod LSTM and LSTM-PF  were discussed in this paper. Based on example of Qiansimen Bridge cable-stayed bridge, it is shown LSTM-PF can be used to predict their state during application of traffic. Bur their parameters are important to collect, can you discuss these parameters you used in this paper. 

Reviewer 2 Report

The topic of the manuscript belongs to the field of the reliability evaluation of online platform data for structural safety. The authors propose a novel temperature compensation method based on long-short term memory network and particle filter to separate the temperature effect from long-term structural health monitoring data.

The work is of great interest to those involved in monitoring large structures, and is proposed for publication in “Remote Sensing

in Structural Health Monitoring” with a few minor corrections.

Line 55 replace “ration” with “ratio”

Line 56 replace “temperate” with “temperature”

Line 113 replace “is then recomputed” with “are then recomputed”

Check Figure 1 the term in the legend “ Hidden state” is missing

Line 157 replace “Matchiing” with “Matching”

Line 159 - 160 replace check the sentence.

Line 186 replace “the” with “The”

Line 187 delete “are”

Line 188 replace “represent” with “represented”

Lines 189-19a replace “represent” with “represented”

Line 190 replace “represent” with “representes”

Line 199 replace “for” with “For”

Line 402 replace “Figure9” with “Figure 9”

Line 451 check the figure 13 description: there are no asterisks and the LSTM-PF prediction is marked as a line with triangles

Table 4 measurement units are missing for absolute values

Line 470 as before check the figure 14, LSTM-PF prediction is marked as a line with triangles

Lines 481, 482, 483 measurement units are missing

Table 5 measurement units are missing

Reviewer 3 Report

This is an important topic to explore and the authors should be congratulated for their efforts. Unfortunately the proposed solution does not appear very practical.

The way this manuscript is written appears to be overly complicated. The term 'Remote Cloud Platform' used in the title and in the paper is somewhat confusing. Basically you are remotely monitoring transportation infrastructure.

It is not clear that the proposed approach is superior to applying some statistical based process control parameters.

The manuscript does not convey deep understanding of bridge behavior, especially a complex suspension bridge. The numerical example presented in Section 4 has no relevance to a bridge. While the authors properly categorize the environmental factors as having a dominant influence, they focus just on the temperature. Significantly, the example provided uses just one temperature location.

One of the problems with analyzing temperature effects is that the bridge is not entirely at the same temperature.  Typically there is a temperature gradient across the entire structure. The problem is how to account for it in a practical fashion. Also, there is an influence of solar radiation. A bridge at 20 degrees C on a sunny day will deflect differently than on a cloudy day.

Speaking of environmental factors as related to a suspension bridge, one of the most significant inputs is the wind load. I suspect that some of the spikes presented in Figure 12 are wind induced.

The authors are correct in concluding that further study is needed. The problem that we are trying to solve is how to devise an early warning system to detect abnormal bridge movements that may be attributed to the onset of structural deterioration as opposed to environmental variables.

Reviewer 4 Report

The manuscript is well written and the method used for compensating for the temperature is well explained. However, I have two main concerns regarding the method. First, it won't be justified to call the method a remote sensing method as the sensors used to collect temperature and deflections are contact-based and installed on the bridge structure. Secondly, the authors have only compensated for the deflection values and not the damping and frequency of vibration recorded for condition assessment. These parameters are the most affected due to temperature change. Also, a neural network-based model can only predict the temperature corrections and not measure the temperature. It has to be noted that the outliers outlined in the text are of major concern during the monitoring of a structure. An outlier indicated abnormal behavior during the regular functioning of the structure.  Unless these concerns are addressed properly, the study will hold lacunas.

Other comments on the manuscript are as follows:

1.      Line 157: Matching spelling

2.  Line 341: Kindly specify the six locations at which the deflections were observed.

3.   Line 343: What do the authors mean by a lot of structural response? The sentence is ambiguous and does not specify the expected responses of the structure at different temperatures.

4.      Line 358: results and discussions